# One-shot In-context Part Segmentation

## ABSTRACT

In this paper, we present the One-shot In-context Part Segmentation (OIParts) framework, designed to tackle the challenges of part segmentation by leveraging visual foundation models (VFMs). Existing training-based one-shot part segmentation methods that utilize VFMs encounter difficulties when faced with scenarios where the one-shot image and test image exhibit significant variance in appearance and perspective, or when the object in the test image is partially visible. We argue that training on the one-shot example often leads to overfitting, thereby compromising the model's generalization capability. Our framework offers a novel approach to part segmentation that is training-free, flexible, and data-efficient, requiring only a single in-context example for precise segmentation with superior generalization ability. By thoroughly exploring the complementary strengths of VFMs, specifically DINOv2 and Stable Diffusion, we introduce an adaptive channel selection approach by minimizing the intra-class distance for better exploiting these two features, thereby enhancing the discriminatory power of the extracted features for the fine-grained parts. We have achieved remarkable segmentation performance across diverse object categories. The OIParts framework not only eliminates the need for extensive labeled data but also demonstrates superior generalization ability. Through comprehensive experimentation on three benchmark datasets, we have demonstrated the superiority of our proposed method over existing part segmentation approaches in one-shot settings.

## CCS CONCEPTS

• **Computing methodologies → Image segmentation**.

## KEYWORDS

Part Segmentation, One-shot Segmentation, Semantic Segmentation

## 1 INTRODUCTION

Part segmentation involves segmenting objects into their constituent parts, providing a more granular understanding of their intricate structure. This granular understanding holds immense potential in various applications, including image editing, object manipulation, and behavior analysis. The task of part segmentation is highly complex, primarily due to the diverse definitions of parts across different object categories and the varying granularity of parts defined for different purposes. Additionally, obtaining an adequate amount of labeled data for this task is both costly and labor-intensive, further

*ACM MM, 2024, Melbourne, Australia*

© 2024 Copyright held by the owner/author(s). Publication rights licensed to ACM.
ACM ISBN 978-x-xxxx-xxxx-x/YY/MM
https://doi.org/10.1145/nnnnnnn.nnnnnnn

increasing the challenge. Therefore, it is crucial to investigate a generalized and data-efficient approach for part segmentation that can flexibly adapt to various objects.

Recent advancements in visual foundation models (VFMs) have revolutionized several computer vision tasks, demonstrating remarkable capabilities across a range of tasks. These models exhibit a remarkable generalization capacity for in-context learning, making them well-suited for adapting to downstream tasks with just a few examples. Efforts such as SegGPT [42] focused on developing generalized in-context learning frameworks for semantic segmentation, enabling inference for novel objects with one labeled example. However, these methods still heavily rely on labeled data for training. Some previous works have primarily focused on exclusively exploring the capabilities of specific VFMs for part segmentation. For instance, SLiMe [14] leveraged the Stable Diffusion [32] model to localize part regions by learning a prompt embedding for each part from only one or a few annotated examples. Nonetheless, these methods still rely on training from a single labeled example, which can result in overfitting and undermine the generalization capabilities of models. Consequently, they encounter challenges in dealing with significant appearance and perspective differences between the test and training examples, as well as difficulties with when the object in the test image is partially visible. These limitations have motivated us to explore a training-free paradigm that relies on a single in-context example for precise part segmentation with superior generalization ability, as the examples shown in Figure 1.

In this paper, we introduce a training-free One-shot In-context Part Segmentation (OIParts) framework designed to unleash the full potential of VFMs in part segmentation, which is achieved by establishing correspondence between an in-context example and the test image leveraging the representations extracted from VFMs. To enhance the representation for fine-grained object parts, we leverage the complementary strengths of two distinct VFMs: DINOv2 and Stable Diffusion. DINOv2 effectively captures dense visual descriptors critical for precise part correspondence, while Stable Diffusion is perceptual to global object structural information. The integration of these two types of features gives rise to two key considerations: **(1) Distinguishability:** The exploration of effective feature fusion techniques to extract discriminative information for fine-grained part segmentation, relying solely on a single example. **(2) Generalization:** It involves developing a harmonious fusion approach for these inherently different-dimensional and -scale features, ensuring that the resulting representation maintains its generalization ability across diverse scenarios.

To address the challenges, we introduce a novel adaptive channel selection approach that minimizes the intra-class distance. We employ this approach to create a distinctive representation for each object part category by selecting channels that improve intra-class compactness, leveraging the information provided by the in-context example. This approach allows us to selectively fuse features from both DINOv2 and Stable Diffusion, yielding a unique representation for each part category that enhances discriminatory power and

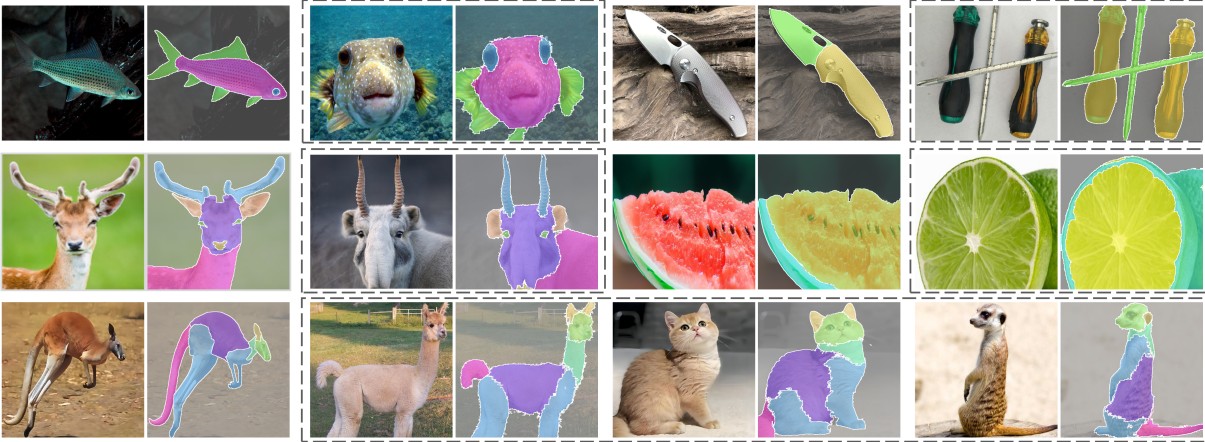

**Figure 1: Part segmentation results in various scenarios. Each in-context example is displayed on the left, with the part segmentation results generated by our OIParts highlighted in the dotted boxes.**

maintains the generalization capabilities of the extracted features. By leveraging this selectively fused feature, we can accurately segment fine-grained object parts by computing the pixel-wise similarity between the provided in-context example and the query image, without requiring extensive labeled data for training. Furthermore, our approach enables flexible and effective selection of the most relevant features for object parts from different in-context examples, offering adaptation for various objects. Hence, our framework exhibits characteristics of generalization ability, data efficiency, and adaptation. The primary contributions of this work are summarized as follows:

- We comprehensively explore the complementary features of DINOv2 and Stable Diffusion to enhance part segmentation effectively, resulting in a training-free framework for one-shot in-context part segmentation by synergizing the complementary strengths of the two models.
- We propose a novel adaptive channel selection approach by improving intra-class compactness to effectively fuse the two features, yielding more discriminative fine-grained part representations without compromising the generalization ability.
- Through comprehensive experimentation, we demonstrate that the segmentation performance of our proposed method surpasses that of existing part segmentation methods utilizing only one in-context example. This superiority is especially notable in datasets with significant pose and perspective variations, such as the horse and car datasets.

## 2 RELATED WORK

### 2.1 Visual Foundation Models

Visual Foundation Models (VFMs) are trained on broad data that can be adapted to a wide range of downstream tasks. One of the key strengths of VFMs is their adaptability and versatility. Unlike traditional models that are often tailored to specific tasks, VFMs exhibit a remarkable ability to generalize. This versatility allows them to be fine-tuned or adapted for different downstream tasks

without the need for extensive retraining or modification. The existing Visual Foundation Models (VFMs) primarily fall into two categories: (1) General Visual Foundation Models: These models learn comprehensive visual representations, forming the foundation for a diverse array of downstream computer vision tasks. They often leverage techniques like self-supervised learning to extract valuable features without heavy reliance on labeled data. Prominent examples include CLIP [29], which excels in zero-shot image recognition by learning from a vast corpus of image-text pairs. Subsequently, several works like [48] have exploited it for few-shot learning. DINO [4], DINOv2 [27], SimCLR [7], MAE [11] and MoCo [12] are other notable models in this category, focusing on contrastive learning to derive robust representations. These models demonstrate remarkable versatility, adapting seamlessly to various vision tasks such as classification, detection, and segmentation. (2) Specialized Vision Foundation Models: These models are tailored to address specific sets of vision problems or tasks. They often exhibit exceptional performance in their respective domains due to their targeted design. For instance, DALL-E [31], DALL-E 2 [30], Stable Diffusion [32], and Imagen [34] are renowned for their proficiency in generating realistic and high-fidelity visual content from textual descriptions. In addition, models like GLIP [19] are specifically crafted for open-set object detection, excelling in identifying objects beyond predefined categories. The recently introduced "segment anything" model (SAM) [15] has garnered significant attention for its remarkable ability to segment objects based on diverse input prompts. Matcher [23] and PerSAM [45] have explored data-efficient semantic segmentation based on SAM. Hummingbird [1] is developed for in-context scene understanding. In this paper, we focus on exploring the pre-trained stable diffusion model and DINOv2 to develop a training-free framework for accurate one-shot part segmentation.

### 2.2 Part Segmentation

Part segmentation, a fundamental task in computer vision, involves the delineation of objects into their constituent parts, thereby providing a more detailed understanding of their intricate structure. As

a fine-grained variety of semantic segmentation, part segmentation has experienced notable advancements parallel to the rapid expansion of semantic segmentation [5, 6, 9, 13, 21, 24]. Previous efforts were predominantly centered around the design and refinement of network architectures. These efforts [10, 20, 25, 39, 40] involved enhancing existing semantic segmentation networks through the integration of novel modules aimed at enhancing contextual information or fine-grained details. Furthermore, some methods [22, 26, 33, 47] explored multi-task joint learning, like edge detection, for utilizing supplementary information from complementary tasks. In addition, Pan *et al.* [28] explored a new open-set part segmentation framework, achieving category-agnostic part segmentation by disregarding part category labels during training. To achieve data-efficient part segmentation, several approaches have explored the use of generative models [2, 37, 46]. Some approaches for universal semantic segmentation often encompass part segmentation as well. These methods, such as SEEM [49], SegGPT [42], Semantic-SAM [18], and HIPIE [41], aim to integrate various semantic segmentation-related tasks into a unified framework, thereby designing a general framework applicable to all segmentation tasks. However, those approaches still heavily rely on extensive labeled data for training. Recently, there have been attempts to utilize VFMs for open-vocabulary or data-efficient part segmentation. For instance, Tang *et al.* [36] proposed a language-driven segmentation model that achieves part segmentation through interactive segmentation. OV-PARTS [43] addressed the issue of data scarcity in open-vocabulary part semantic tasks by introducing two open-vocabulary datasets. They also explored utilizing VFMs to assist in open-vocabulary part segmentation. In addition, Sun *et al.* [35] designed an open-vocabulary part segmentation algorithm combined with object detection, aiming to simultaneously address the issues of open object categories and open part categories in part segmentation, and utilized the DINOv2 model for visual part features extraction. SLiMe [14] introduced a method aimed at part segmentation with arbitrary granularities, which was achieved by harnessing the text and visual features alignment power of attention mechanisms inherent in the diffusion generation process of Stable Diffusion. Unlike existing training-based methods, we introduce a training-free approach that combines the complementary strengths of two distinct VFMs. Furthermore, we comprehensively explore the fusion of these two feature types to achieve more accurate and generalized one-shot in-context part segmentation.

## 3 METHODS

We present a novel One-shot In-context Part Segmentation (OIParts) framework designed to achieve part segmentation with just one labeled image as an in-context example, leveraging existing visual foundation models without requiring any training or fine-tuning. Given an in-context example composed of an image $I_r \in \mathbb{R}^{H \times W \times 3}$ and a corresponding binary mask $M_r \in \mathbb{R}^{H \times W \times C}$, OIParts can segment the object in query image $I_q$ into desired parts as defined in $M_r$, where $M_r$ denotes binary mask of $C$ object parts. The overview of the whole framework is illustrated in Figure. 2. Specifically, we employ the pre-trained stable diffusion model (SD) and DINOv2 to extract complementary semantic features for images $I_r$ and $I_q$. Further, we fuse these two types of features with an adaptive channel

selection mechanism to obtain more distinctive representations for each part. Subsequently, the selectively fused features are used to calculate the pixel-wise semantic similarity between $I_r$ and $I_q$, we can obtain segmentation masks by transferring the pixel-wise label in the in-context example $M_r$ to the novel query images guided by the computed semantic similarity. In the following subsections, we will delve into the details of the overall process.

### 3.1 Feature Extraction

Previous works have demonstrated DINOv2's capability to provide explicit information crucial for semantic segmentation tasks. Additionally, Stable Diffusion (SD) exhibits a robust internal representation of objects, effectively capturing both their content and layout. Leveraging these strengths, we employ DINOv2 to extract dense visual descriptors for object parts and utilize SD to derive complementary global structure information, thereby enhancing the overall part representation. Specifically, we extract the token features from layer 11 of DINOv2 for each image, denoted as $F^{dino}$, and extract the SD features $F^{sd}$ from the denoising U-Net.

To align the scales and distributions of these two types of features, we first normalize the SD feature and DINOv2 feature by L2 normalization respectively following [44]. Then concatenate them along the channel dimension to get the feature $F \in \mathbb{R}^{H' \times W' \times D}$:

$$F = \text{Concat}(\|F^{sd}\|_2, \|F^{dino}\|_2) \tag{1}$$

### 3.2 Adaptive Channel Selection

Considering that not every channel of the feature contributes meaningful information for each object part, we perform feature selection for the concatenated features. Some channels may be corrupted by noise or may capture irrelevant variations, thereby obscuring the distinctiveness of the representation for the specific part. Therefore, it becomes crucial to eliminate these noisy channels for specific object parts, ensuring a more discriminative and focused feature representation. Although it may seem intuitive to adopt a learning-based method for further fusing the two types of features, relying solely on a one-shot in-context example could undermine their generalization capabilities, as explored in the experiments section. Instead, we present an innovative approach leveraging channel selection to generate more discriminative representations using the two complementary features without additional training.

To achieve this, two primary concerns need to be addressed: (1) Identifying the specific channels that should be chosen to achieve the desired discriminative power; and (2) Determining the optimal number of channels to be selected. In this paper, we introduce a novel adaptive channel selection mechanism by formulating an optimization problem that minimizes the intra-class distance. By solving this optimization problem, we aim to select discriminative channels that effectively capture the distinguishing characteristics of each part. This selection process ensures that the new features formed by the selected channels minimize the distance within the same class, allowing for the effective separation of different classes.

To identify representative feature channels for each object part, we select channels that generate new features, thereby enhancing the compactness of data points associated with the same object part in the new feature space. This approach can implicitly improve the separability of data points from different object parts, as illustrated

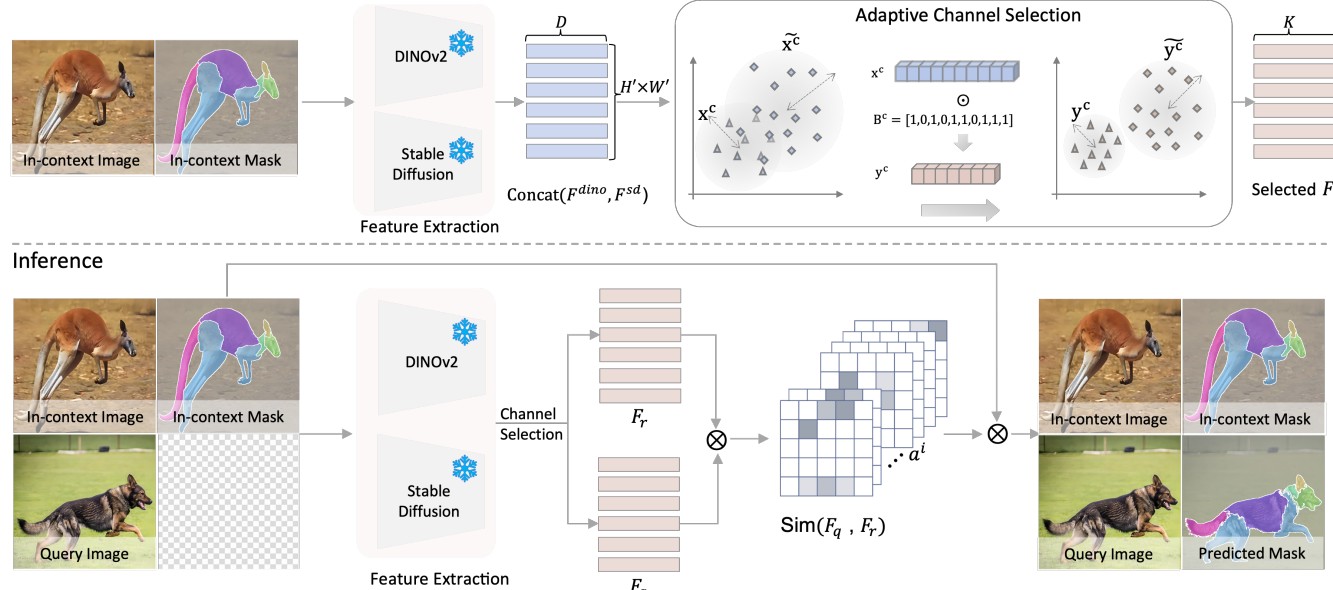

**Figure 2: The overall framework of our proposed OIParts. We acquire features for each image by extracting them from DINOv2 and SD. Initially, we calculate representative feature channels for each object part based on the provided in-context example. During the inference phase, we generate representative features for each object part. Subsequently, cosine similarity scores are calculated between pixels of the in-context image and the query image, which are further utilized to predict part masks for the query image.**

in Figure 2. Specifically, given the L2 normalized feature $F_r$ of the in-context example, we denote the pixels in $F_r \in \mathbb{R}^{H' \times W' \times D}$ corresponding to part $c$ as $x^c = \{x_i^c \in \mathbb{R}^D, i = 1, \ldots, |M_r^c|\}$, where $M_r^c$ represents the binary mask corresponding to part $c$, and $|M_r^c|$ is the number of pixels belonging to the part $c$ at the resolution $H' \times W'$. Conversely, $\widetilde{x^c}$ represents pixels that do not belong to part $c$. We use a binary matrix $B \in \{0, 1\}^{C \times D}$ to indicate the selected channels for each category. Therefore, $B^c \in \{0, 1\}^D$ is a binary vector that denotes whether each channel is chosen for part category $c$. Then, to select $K$ representative channels, we define the channel selection as an optimization problem. The objective of this optimization is to minimize the intra-class distance computed using the new feature vectors composed of $K$ selected channels:

$$\min \mathcal{D}(x^c \odot B^c) + \mathcal{D}(\widetilde{x^c} \odot B^c), \text{ s.t. } B^c (B^c)^\top = K, \quad (2)$$

where $x^c \odot B^c$ denotes only selecting the feature channels according to $B^c$, $\mathcal{D}(\cdot)$ is used to measure the distance of the feature set, $K$ is the number of channels to be selected.

Hence, our goal is to identify a subset of K channels. When these channels represent a pixel, they minimize the distance between pixels belonging to the same part. Here, we utilize a straightforward metric, variance, for $\mathcal{D}(\cdot)$ to identify the optimal subset among the various subsets of K channels. Although we have explored several metrics, like Kullback-Leibler divergence and Jensen–Shannon divergence, as evaluated in the experimental section, we have found that adopting variance is the simple yet most effective approach. We denote the feature set $x^c \odot B^c$ as $y^c = \{y_i^c \in \mathbb{R}^K, i = 1, \ldots, |M_r^c|\}$,

and $\overline{y_j^c}$ denotes the mean of $j$-th channel. Thus, $\mathcal{D}(\cdot)$ can be formulated as:

$$\mathcal{D}(x^c \odot B^c) = \frac{1}{K} \sum_j^K \frac{1}{|M_r^c|} \sum_i^{|M_r^c|} (y_{ij}^c - \overline{y_j^c})^2 \quad (3)$$

Given that the variance of a set of feature vectors is calculated separately for each channel, we can efficiently solve this optimization problem of Equation 2 to obtain $B^c$ by ranking the variances of all $D$ channels and subsequently selecting the top $K$ channels with the lowest variances. Those $K$ channels form a subset with the minimum variance among all subsets of $K$ channels. This approach is also intuitively reasonable, as a channel with low variance indicates that it represents a common characteristic among pixels belonging to that part, thereby making it suitable as the representative channel for that part.

We further evaluate the segmentation accuracy on the given in-context example to find the optimal value of $K$ for each part category $c$. Specifically, given a particular value of $K$, we utilize the aforementioned method to decide which channels should be selected for the part category. Subsequently, we compute two class centers corresponding to two feature vector sets, one belonging to part $c$ and the other not. These two class centers are then utilized to re-assign labels for the in-context example to obtain a clustered mask $\hat{M}_r^c$. By varying the value of $K$, we can obtain different clustered masks $\hat{M}_r^c$. Finally, we evaluate the accuracy of the clustered mask $\hat{M}_r^c$ with the corresponding ground-truth mask $M_r^c$, choosing the value of $K$ that yields the highest accuracy for our approach.

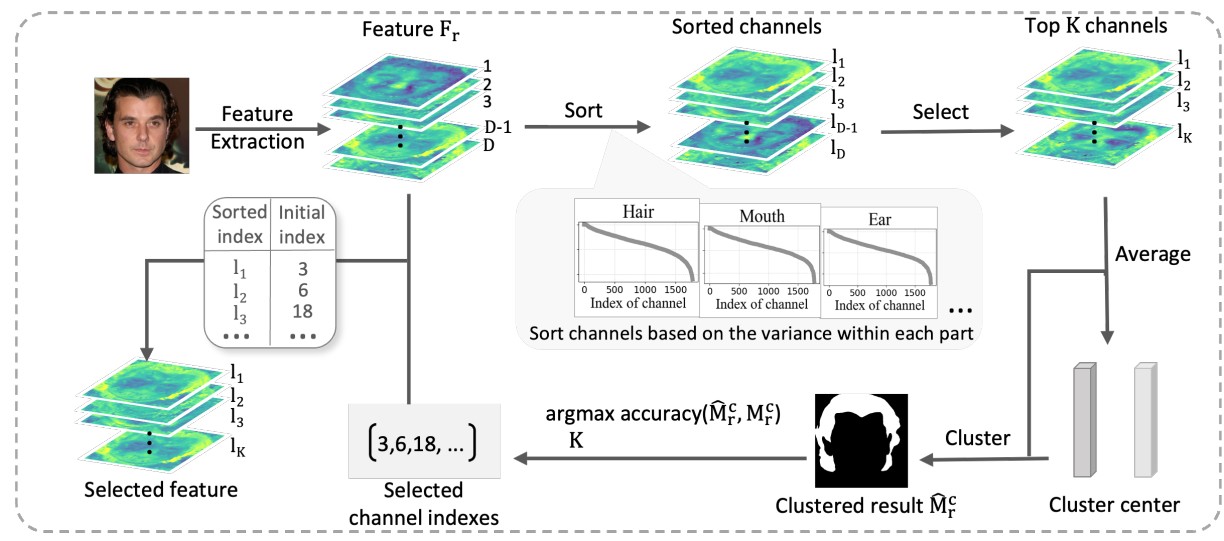

**Figure 3: The overall pipeline of the proposed channel selection.**

The overall pipeline of performing this channel selection is illustrated in Figure 3. Note that this channel selection process only needs to be calculated once using the in-context example and can subsequently be applied during inference.

## 3.3 Part Segmentation

To perform part segmentation for a query image $I_q$ using the in-context example, we first extract the complementary feature and utilize $B$ to obtain the selectively fused feature $F_q \in \mathbb{R}^{H' \times W' \times K}$ for each part category. Subsequently, we use cosine similarity to measure the pixel-wise semantic similarity. For each pixel in $F_q^i$, we can calculate the similarity score with each pixel in $F_r^j$ by:

$$s^{i,j} = \frac{1}{\beta} \frac{\langle F_q^i, F_r^j \rangle}{\|F_q^i\| \times \|F_r^j\|}, a^i = \underset{j}{\text{softmax}}(s^i), \qquad (4)$$

where $S^{i,j}$ denotes the cosine similarity between $F_r^i$ and $F_q^i$, $beta$ serves as a hyper-parameter for scaling value before applying a softmax operation to obtain the score values, and $a^{i,j}$ stands for the similarity score that have been normalized using the softmax function. We then utilize this similarity score to combine the corresponding labels from $M_r$, thereby generating a prediction for each pixel in the query image and acquiring the predicted part segmentation mask $M_q^c$:

$$M_q^c = \sum_j a^{i,j} M_r^{c,j} \qquad (5)$$

Finally, by concatenating the predictions of all the parts, we obtain the final part segmentation prediction $M_q$. This prediction is further upsampled to the original image size using bilinear interpolation. In addition, the resolution of the extracted features used for part segmentation is relatively low, resulting in a loss of object details and coarse segmentation results around the boundary regions. To mitigate this issue and enhance the quality of segmentation, we incorporate an edge smoothing technique called the Fast

Bilateral Solver (FBS) [3]. This technique effectively refines the coarse segmentation masks, providing more precise boundaries, thereby improving the overall accuracy and visual quality of the segmentation results.

## 4 EXPERIMENTS

In this section, we comprehensively evaluate our approach both qualitatively and quantitatively.

### 4.1 Experimental Settings

**Implementation Details** In our experiments, we employ the DINOv2 [27] and Stable Diffusion v1-5 model [32] for feature extraction following [44]. For the Stable Diffusion model, we set the timestep to 100 and use a generic text prompt template like "a photo of $c$", where $c$ is the corresponding category name. The feature maps extracted from both the Stable Diffusion and DINOv2 models are at a consistent resolution of $60 \times 60$ with dimensions of 768 and 1024, respectively. All experiments are conducted on NVIDIA RTX3090 GPU.

**Datasets and Metrics** We conduct experiments on two datasets of three distinct object categories, PASCAL-Part [8] and CelebAMask-HQ [17], following the same dataset setting as SLiMe [14]. We evaluate our results using the mean Intersection over Union (mIoU) metric. PASCAL-Part provides comprehensive annotations of various object parts across images, encompassing 20 distinct object categories. We focus on the object categories of *car* and *horse*. In the *car* category, the object is annotated into six parts: *body*, *light*, *plate*, *wheel*, *window* and *background*. In the *horse* category, the object is annotated into five parts: *head*, *neck+torso*, *legs*, *tail*, and *background*. CelebAMask-HQ is a large-scale face image dataset created for facial segmentation tasks. We report results on the parts used in ReGAN and SLiMe for comparison, which divide the face into ten parts: *cloth*, *ear*, *eye*, *eyebrow*, *skin*, *hair*, *mouth*, *neck*, *nose* and *background*.

**Table 1: Comparison to other 1-shot and 10-shot methods on the face dataset.**

| Part Name | 10-shot | 1-shot | | | |
|---|---|---|---|---|---|
| | *ReGAN* | *SegGPT* | *SegDDPM* | *SLiMe* | *Ours* |
| Cloth | 15.5 | 24.0 | 28.9 | 52.6 ± 1.4 | 60.9 |
| Brow | 68.2 | 48.8 | 46.6 | 44.2 ± 2.1 | 48.1 |
| Ear | 37.3 | 32.3 | 57.3 | 57.1 ± 3.6 | 63.4 |
| Eye | 75.4 | 51.7 | 61.5 | 61.3 ± 4.6 | 65.0 |
| Hair | 84.0 | 82.7 | 72.3 | 80.9 ± 0.5 | 82.2 |
| Mouth | 86.5 | 66.7 | 44.0 | 74.8 ± 2.9 | 79.1 |
| Neck | 80.3 | 77.3 | 66.6 | 78.9 ± 1.3 | 76.9 |
| Nose | 84.6 | 73.6 | 69.4 | 77.5 ± 1.8 | 74.0 |
| Skin | 90.0 | 85.7 | 77.5 | 86.8 ± 0.3 | 86.5 |
| BG | 84.7 | 28.0 | 76.6 | 81.6 ± 0.8 | 83.8 |
| mIoU | 70.7 | 57.1 | 60.1 | 69.6 ± 0.3 | **72.0** |

## 4.2 Quantitative Comparisons

We conduct experiments on the same test sets of the mentioned datasets employed in previous methods to ensure fair comparisons. We mainly compare our method with three existing methods across three distinct categories of datasets. In the 1-shot setting, our primary comparisons are with SegGPT, SegDDPM [2] and SLiMe. SegGPT explored an in-context learning framework for semantic segmentation training on a vast amount of annotated data, enabling inference with just one in-context example. SegDDPM explored denoising diffusion probabilistic models as an effective source of image representation for semantic segmentation. SLiMe was designed for part segmentation, exploiting the SD model capable of learning a part prompt using just one labeled example. Notably, our method even surpasses other methods in the 10-shot settings and even some fully supervised part segmentation methods [38, 39]. For a more comprehensive comparison, we also include ReGAN in the 10-shot settings. ReGAN leverages pretrained GAN models, specifically trained on the FFHQ and LSUN-Horse datasets for face and horse part segmentation. Additionally, for car part segmentation, ReGAN employs a pre-trained GAN from the LSUN-Car dataset.

**Comparison on the Face Dataset** The results presented in Table 1 demonstrate the effectiveness of our method compared to other 1-shot and 10-shot approaches on the CelebAMask-HQ10 dataset. Overall, our method surpasses SLiMe, SegDDPM and Seg-GPT in terms of mIoU performance and for the majority of facial parts in the 1-shot setting, achieving a mIoU of 72.0% compared to 69.6% of SLiMe, 60.1% of SegDDPM and 57.1% of SegGPT. Notably, our method achieves these results without any training or fine-tuning, whereas SegGPT requires a large annotated dataset for supervision, and SLiMe necessitates specific fine-tuning. Additionally, despite the inherent disadvantage of comparing against a 10-shot method like ReGAN, our approach still outperforms Re-GAN on mIoU, achieving 1.3% improvements compared to 70.7% of ReGAN. It's worth noting that the comparisons made here highlight the robustness and effectiveness of our method, particularly in scenarios where annotated data is limited or fine-tuning is impractical.

**Comparison on the Car Dataset** Car images in PASCAL-Part present distinct challenges compared to well-aligned face images

**Table 2: Comparison to other 1-shot and 10-shot methods on the car dataset.**

| Part Name | Supervised | | 10-shot | 1-shot | | |
|---|---|---|---|---|---|---|
| | *CNN* | *CNN+CRF* | *ReGAN* | *SegGPT* | *SLiMe* | *Ours* |
| Body | 73.4 | 75.4 | 75.5 | 62.7 | 79.6 ± 0.4 | 77.7 |
| Light | 42.2 | 36.1 | 29.3 | 18.5 | 37.5 ± 5.4 | 59.1 |
| Plate | 41.7 | 35.8 | 17.8 | 25.8 | 46.5 ± 2.6 | 57.2 |
| Wheel | 66.3 | 64.3 | 57.2 | 65.8 | 65.0 ± 1.4 | 66.9 |
| Window | 61.0 | 61.8 | 62.4 | 69.5 | 65.6 ± 1.6 | 59.2 |
| BG | 67.4 | 68.7 | 70.7 | 77.7 | 75.7 ± 3.1 | 71.1 |
| mIoU | 58.7 | 57.0 | 52.2 | 53.3 | 61.6 ± 0.5 | **65.2** |

**Table 3: Comparison to other 1-shot and 10-shot methods on the horse dataset.**

| Part Name | Supervised | | 10-shot | 1-shot | | | |
|---|---|---|---|---|---|---|---|
| | *Shape+ Apperence* | *CNN+ CRF* | *ReGAN* | *Seg GPT* | *Seg DDPM* | *SLiMe* | *Ours* |
| Head | 47.2 | 55.0 | 50.1 | 41.1 | 12.1 | 61.5 ± 1.0 | 73.0 |
| Leg | 38.2 | 46.8 | 49.6 | 49.8 | 42.4 | 50.3 ± 0.7 | 50.7 |
| Neck+Torso | 66.7 | - | 70.5 | 58.6 | 54.5 | 55.7 ± 1.1 | 72.6 |
| Tail | - | 37.2 | 19.9 | 15.5 | 32.0 | 40.1 ± 2.9 | 60.3 |
| BG | - | 76.0 | 81.6 | 36.4 | 74.1 | 74.4 ± 0.6 | 77.7 |
| mIoU | - | - | 54.3 | 40.3 | 43.0 | 56.4 ± 0.8 | **66.9** |

in CelebA-HQ10, as they exhibit larger variations in perspective and appearance. Table 2 presents the results for the car class. In the 1-shot setting, our method outperforms SegGPT and SLiMe in terms of mIoU, yielding improvements of 11.9 (53.3% *vs.* 65.2%) and 3.6% (61.6% *vs.* 65.2%) , respectively. Qualitative results are illustrated in Figure 2, demonstrating the superior performance of our method compared to SegGPT and SLiMe. In the 10-shot setting, our method outperforms ReGAN in terms of mIoU. Additionally, our method performs better than fully supervised baselines like CNN [38] and CNN+CRF [16, 38].

**Comparison on the Horse Dataset** The part segmentation of horse images in PASCAL-Part is more challenging than the other two object categories because of the ambiguity in distinguishing between different parts and the horse object in this dataset is usually partially visible. Table 3 shows our results on the horse class. Our method exhibits superior performance. For the 1-shot setting, our method has a large improvement over SegGPT, SegDDPM and SLiMe in all parts as well as on mIoU, gaining improvements of 26.6% (40.3% *vs.* 66.9%), 23.9% (43.0% *vs.* 66.9%) and 10.5% (56.4% *vs.* 66.9%), respectively. Furthermore, our method outperforms ReGAN on mIoU in the 10-shot setting. For fully supervised baselines like shape+Apperence [39] and CNN+CRF [16, 38] our method also performs much better in the report results.

## 4.3 Qualitative Comparisons

To gain a deeper understanding of our method's performance, we conduct a qualitative comparison with SLiMe and SegGPT. As depicted in Figure 4, in the face dataset example, SLiMe often produces

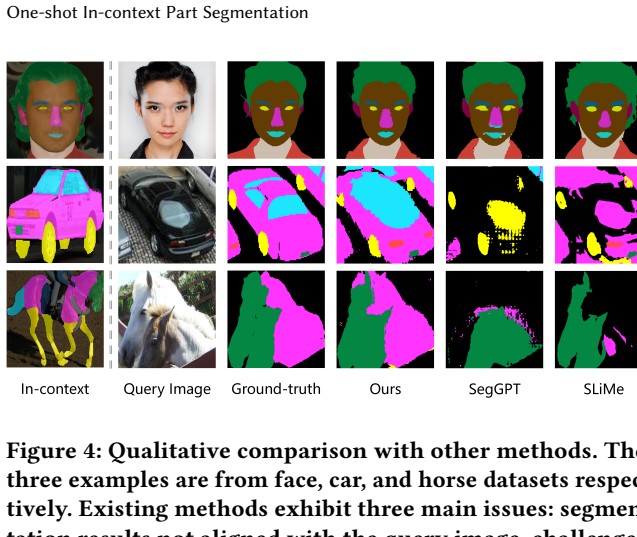

Figure 4: Qualitative comparison with other methods. The three examples are from face, car, and horse datasets respectively. Existing methods exhibit three main issues: segmentation results not aligned with the query image, challenges in handling perspective differences, and difficulty with partially visible objects.

erroneous segmentations that do not align with the original image, particularly near the hair area. SegGPT tends to produce noisy segmentations, especially around the nose and mouth regions. In contrast, our method excels in accurately capturing fine-grained object details and delineating object parts, surpassing existing methods in this aspect. From the example of the car dataset, when the in-context example and the query image are captured from different perspectives, the performance of SLiMe and SegGPT methods can be significantly affected. As demonstrated in Figure 4, when the query image is taken from an aerial perspective, while the provided example is from a distinctly different perspective, both SegGPT and SLiMe struggle to achieve accurate part segmentation. From the example of the horse dataset, we observe that when the object in the query image is partially visible, segmentation performance suffers, indicating a challenge for SegGPT and SLiMe in handling occluded or partially visible objects.

Additionally, given that both SLiMe and our proposed method rely solely on one labeled example, we conducted further experiments to comprehensively compare with SLiMe. As illustrated in Figure 5, we conduct additional experiments to evaluate the segmentation of a single query image using various in-context examples. Our observations reveal that SLiMe's performance on the same query image fluctuates considerably depending on the provided examples. Specifically, SLiMe tends to perform well when the in-context example closely resembles the query image, but its performance deteriorates rapidly when significant differences exist between them. This pattern can be traced back to SLiMe's reliance on training specifically with the in-context example, which may result in overfitting to that specific example. Although our method also exhibits fluctuations, we demonstrate better stability compared to SLiMe across various in-context examples.

## 4.4 Ablation Study

In this section, to comprehensively evaluate the contributions of different components in our approach, we conducted an ablation study on the face and car dataset as presented in Table 4. Each row in

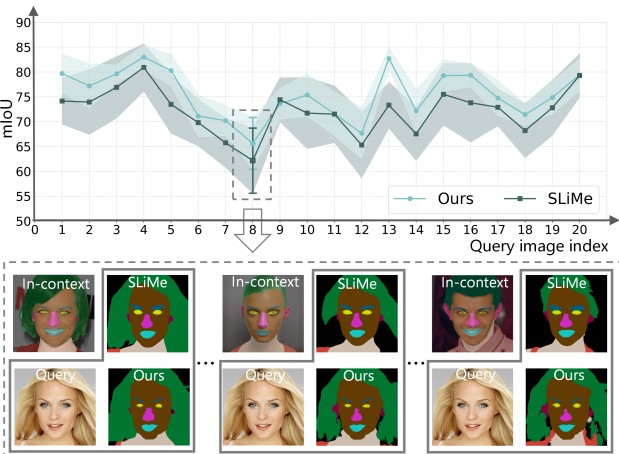

Figure 5: Comparison between our proposed method and SLiMe across various in-context examples. Evaluations were performed on 20 randomly selected query images using 8 distinct randomly selected in-context examples.

Table 4: Ablation study results. The contributions of various components in our approach.

| DINOv2 | SD | Selection | FBS | Car | Face |
|--------|-----|-----------|-----|------|------|
| ✓ | | | | 60.2 | 63.4 |
| | ✓ | | | 39.5 | 62.0 |
| ✓ | ✓ | | | 61.0 | 66.9 |
| ✓ | ✓ | ✓ | | 62.5 | 67.9 |
| ✓ | ✓ | ✓ | ✓ | 65.2 | 72.0 |

the table represents a distinct configuration of components, and the corresponding segmentation performance is reported as the average mIoU score. Through this study, we explored multiple strategies aimed at gradually improving part segmentation performance.

**Discussion of the SD and DINOv2 Features** To demonstrate the complementary nature of SD and DINOv2 features for part segmentation, we conducted experiments using each feature separately. Initially, from Table 6, it is evident that SD outperforms on specific parts, while DINOv2 features excel on others, highlighting their complementary characteristics. Furthermore, simply concatenating the two features results in an improvement in mIoU, with enhancements of 3.5% and 4.9% on the face dataset and 0.8% and 21.5% on the car dataset compared to using DINOv2 and SD features individually, as depicted in Table 4. Given that DINOv2 excels at capturing dense descriptors for local matching, while SD excels at perceiving global structures, their combination leverages the strengths of both features.

**Effectiveness of the Channel Selection** To validate the effectiveness of the proposed adaptive channel selection approach, we contrast it with the concatenation operation without further selection. As shown in Table 4, our channel selection approach yields improvements of 1.5% (61.0% *vs.* 62.5%) and 1.0% (66.9% *vs.* 67.9%) on the car and face datasets, respectively. Additionally, Table 6 reveals

**Table 5: Evaluation of different distance metrics for the proposed feature selection.**

|  | w/o Selection | Variance | Cosine | KL | JS |
|---|---|---|---|---|---|
| mIoU | 66.9 | 67.9 | 67.0 | 67.7 | 67.3 |

improvements across nearly all parts, indicating the capability of our adaptive channel selection approach to generate more representative features for each part. This enhancement facilitates one-shot part segmentation. We further provide a visualized example with and without the channel selection in Figure 6. We can observe that without the channel selection, the hand area is misclassified as face skin and neck, whereas with our channel selection, it is correctly classified as background, well demonstrating the effectiveness of our approach.

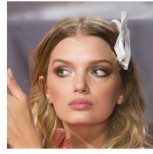 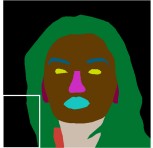 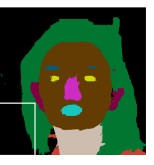 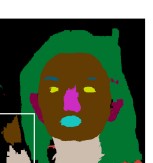

| Image | Ground-truth | Selection | w/o Selection |

**Figure 6: Qualitative comparison of channel selection.**

**Discussion of the Distance Metrics for Channel Selection** We explore several metrics used in Equation 2 to identify the channels to be selected. In addition to variance, we further investigate cosine distance, Kullback-Leibler (KL) divergence, and Jensen–Shannon (JS) divergence. The results are presented in Table 5. Both of these distance metrics are capable of selecting representative features, thereby enhancing the mIoU. However, the straightforward application of variance enables us to identify the channels that achieve optimal performance, resulting in the highest mIoU of 67.9%. Channels with low variance tend to have more consistent feature values within the part, indicating that they may capture more relevant information for distinguishing between different parts of objects. Therefore, we adopt variance as the preferred distance metric for channel selection in our method.

**Selection-based Fusion vs. Learning-based Fusion** To further evaluate the effectiveness of our proposed selection-based feature fusion, we train a linear classifier with two linear projection layers on top of the extracted complementary features of the in-context example. We conduct two additional experiments: first, we exploit the classifier's ability to perform part segmentation directly, and the results are detailed in Table 6 under the column labeled "Classifier". Second, we utilize the intermediate linear projection layer in the classifier to fuse the two features and perform segmentation like our approach as described in Equations 4 and 5, results are reported in Table 6 under the column labeled "Classifier Feature". Our analysis demonstrates that direct part segmentation using the linear classifier achieved a notably low accuracy of 45.4%. When fusing the SD and DINOv2 features using the linear projection layer, the performance improves to 64.3%, yet remains lower than directly concatenating the two features, which yields 66.9%. Notably, our

**Table 6: Comparison of learning-based fusion with our selection-based fusion on the face dataset.**

| Part | Classifier | Classifier Feature | DINOv2 | SD | DINOv2+SD | DINO+SD+ Selection |
|---|---|---|---|---|---|---|
| Cloth | 9.9 | 26.0 | 47.8 | 34.6 | 49.7 | 51.6 |
| Brow | 7.7 | 48.2 | 45.2 | 46.0 | 51.3 | 52.8 |
| Ear | 42.4 | 53.4 | 52.4 | 59.5 | 56.2 | 56.7 |
| Eye | 18.1 | 59.0 | 62.8 | 47.8 | 61.4 | 62.6 |
| Hair | 65.3 | 78.6 | 71.8 | 73.7 | 76.8 | 76.9 |
| Mouth | 28.3 | 67.8 | 58.1 | 58.3 | 66.0 | 69.5 |
| Neck | 53.7 | 71.1 | 67.0 | 71.4 | 70.5 | 71.5 |
| Nose | 71.6 | 75.0 | 74.6 | 71.4 | 75.5 | 75.4 |
| Skin | 78.7 | 86.0 | 82.4 | 82.4 | 84.0 | 84.4 |
| BG | 77.9 | 78.1 | 71.6 | 75.2 | 78.0 | 77.9 |
| mIoU | 45.4 | 64.3 | 63.4 | 62.0 | 66.9 | 67.9 |

selective fusion approach achieved the highest accuracy of 67.9%. These findings underscore the effectiveness of our selection-based feature fusion method, surpassing learning-based fusion techniques. Additionally, our part segmentation method outperforms directly training a classifier based on the extracted features.

**Post-processing** To address the loss of spatial details stemming from the relatively small size of the extracted features, we employ an edge-aware smoothing algorithm FBS, which aids in restoring finer details around boundaries. Results in Table 4 showcase the effectiveness. Nevertheless, it is worth mentioning that these improvements are primarily attributed to the detailed cues provided by the higher-resolution inputs.

**Generalization Capability** The proposed OIParts is a training-free framework, thereby preserving the generalization capabilities of the VFMs. This remarkable feature ensures that our method can be seamlessly applied to various object categories, needing only a single in-context example. As illustrated in Figure 1, with just one labeled example, OIParts precisely segments the parts of novel objects and exhibits robust performance when dealing with objects in different poses or perspectives.

## 5 CONCLUSION

In conclusion, our One-shot In-context Part Segmentation (OIParts) framework presents a pioneering solution to the challenges of part segmentation by harnessing visual foundation models (VFMs). We address the limitations of existing training-based methods, which struggle with variance in appearance, perspective, and partial visibility between one-shot and test images, often leading to overfitting and reduced generalization. Our framework introduces a novel, training-free approach that requires only a single in-context example for precise segmentation with superior generalization ability. Through a comprehensive exploration of VFMs' strengths, particularly DINOv2 and Stable Diffusion, we integrate an adaptive channel selection approach by minimizing intra-class distance, enhancing feature extraction and discriminatory power for fine-grained part segmentation. Our method achieves remarkable performance across diverse object categories, showcasing its effectiveness in one-shot settings.

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
