# OpenReview forum: "One-shot In-context Part Segmentation"
_acmmm.org/ACMMM/2024/Conference — MM2024 Poster_

### Official Review · Reviewer_4CjB · 2024-05-21

**Rating:** 6
**Confidence:** 2

**Summary:**

This paper targets on the task of part segmentation via visual foundation models (VFMs). To relieve the overfitting of existing methods that train on the one-shot example, OIParts, a training-free framework is proposed. With the concatenated feature from DINOv2 and SD, an adaptive channel selection approach is proposed. Through comprehensive experimentation on three benchmark datasets, author have demonstrated the superiority of OIParts over existing part segmentation approaches in one-shot settings.

**Strengths:**

1. The writing is easy to understand.
2. The proposed method sounds reasonable and novel.
3. A bunch of experiments are provided.

**Limitations:**

1. The organization can be improved. For example, L66-88 is too long. I suggest the authors to be more concise in the Intro and leave more details to the related work. Besides, using a single long paragraph for each related work is not a good idea as in L168-274.
2. Does in-context segmentation means one-shot? if yes, whether previous one-shot VOS tasks like [1] belongs to in-context learning?
3. Fig. 3 can be further polished.

Ref:
1. In-N-Out Generative Learning for Dense Unsupervised Video Segmentation.

**Suitability:**

3

---

### Official Review · Reviewer_Zghq · 2024-05-22

**Rating:** 3
**Confidence:** 3

**Summary:**

This paper presents a novel framework for part segmentation tasks using visual foundation models (VFMs). It introduces a training-free, flexible, and data-efficient approach that leverages VFMs to perform part segmentation using only a single in-context example, which is designed to improve generalization and eliminate the need for extensive labeled data. And it demonstrates significant improvements in segmentation accuracy and generalization capabilities across various datasets.

**Strengths:**

1.	The introduction of an adaptive channel selection mechanism to improve intra-class compactness and enhance feature discrimination is novel. By selectively fusing features from both models, the framework achieves more discriminative part representations, which is a significant improvement over existing methods that do not employ such adaptive techniques.
2.	The theoretical underpinning of a training-free framework is a significant strength. By avoiding extensive training and fine-tuning, the approach minimizes overfitting and enhances generalization capabilities. This makes the method highly adaptable and efficient, requiring only a single in-context example for accurate segmentation.
3.	Detailed ablation studies are conducted to validate the contributions of different components of the framework, such as the use of DINOv2 and Stable Diffusion features, adaptive channel selection, and post-processing with Fast Bilateral Solver (FBS). These studies provide a clear understanding of the importance and impact of each component on the overall performance.

**Limitations:**

1.	The paper builds upon existing VFMs such as DINOv2 and Stable Diffusion, and while it combines them in a novel way, the core techniques and models are not entirely new. Previous works like SLiMe and SegGPT have already explored the use of VFMs for segmentation tasks. For example, SLiMe also leverages the Stable Diffusion model for part segmentation, highlighting similar complementary strengths. This could suggest that the paper's novelty primarily lies in the integration and slight modification of existing methods rather than introducing fundamentally new concepts.
2.	While this paper aims to mitigate overfitting by avoiding training on one-shot examples, it does not ensure and prove that adaptive channel selection does not implicitly cause model overfitting through its optimization process. It is crucial, especially when fine-tuning the number of channels or the selection mechanism based on a minimum number of examples.
3.	The experiments are conducted on three datasets, but the diversity of the datasets could be questioned. PASCAL-Part and CelebAMask-HQ are commonly used datasets, but additional evaluation on more varied and challenging datasets could provide a better assessment of the generalization capabilities of the proposed method. Furthermore, the paper primarily compares its method to a limited number of existing approaches.

**Suitability:**

2

---

### Official Review · Reviewer_Fokb · 2024-05-27

**Rating:** 4
**Confidence:** 4

**Summary:**

The paper introduces a novel framework named One-shot In-context Part Segmentation (OIParts) that addresses the challenges of part segmentation using visual foundation models (VFMs) without requiring extensive labeled data. This training-free approach leverages DINOv2 and Stable Diffusion to enhance generalization and segmentation accuracy. The framework employs an adaptive channel selection mechanism by formulating an optimization problem that minimizes the intra-class distance, which can learn more discriminative features of each part. The proposed method achieves superior segmentation performance across various datasets compared to existing methods utilizing only one in-context example.

**Strengths:**

The paper presents a novel training-free approach for part segmentation by proposing an adaptive channel selection method and utilizing the pretraining features of Visual Foundation Models. The paper is well-organized, with clear explanations of the methodology and thorough experimental analysis. Figures and tables effectively illustrate the results and comparisons with baseline methods. The proposed method is technically robust, utilizing state-of-the-art VFMs and demonstrating a clear understanding of the challenges in part segmentation. The experimental results are well-documented and show consistent improvements over existing methods.

**Limitations:**

1、While the combination of DINOv2 and Stable Diffusion is effective, the main innovation lies in the adaptive channel selection. Why not consider different combinations of other Visual Foundation Models？ More combinations of experiments can demonstrate the effectiveness and generality of the proposed method.
2、The proposed method involves complex processes like feature extraction from two different VFMs and an adaptive channel selection mechanism. This could pose challenges for practitioners in terms of implementation and computational resources. Does the author have any plans for open source code？
3、The reviewer is concerned about the scalability of the method. The method's reliance on high-resolution feature extraction and adaptive selection might limit its scalability for very high-resolution images due to computational overhead.
4、If there are several components in the in context image that are outside the screen or are obstructed, how should the proposed method handle this situation?

**Suitability:**

3

---

### Official Review · Reviewer_wFAk · 2024-05-27

**Rating:** 4
**Confidence:** 4

**Summary:**

This paper introduces a training-free, flexible, and data-efficient one-shot in-context part segmentation method, which leverages the complementary strengths of DINOv2 and Stable Diffusion. Extensive experiments conducted on three benchmark datasets showcase the superiority of the proposed approach.

**Strengths:**

The concept of exploring the complementary strengths of DINOv2 and Stable Diffusion for part segmentation is interesting.

**Limitations:**

1、Does the channel selection require the use of a mask? If a mask is required, how is the channel selection performed on the query image?
2、Figure 2 is unclear regarding how the In-Context Mask guides the segmentation of the query image and should be clarified.

**Suitability:**

2

---

### Meta-Review · Area_Chair_W76u · 2024-07-03

**Recommendation:** Accept (Poster)
**Confidence:** 5

**Metareview:**

This paper introduces a part segmentation method by leveraging visual foundation models. The original reviews are one accept, two borderline accept, and one borderline reject. The main concerns center around the mechanism of channel selection and in-context mask guidance. The rebuttal successfully addressed some of the concerns and one reviewer raised the final rating to weak accept.